# Osteoporosis is associated with increased CVD mortality and all-cause mortality in alcohol-consuming individuals: A cohort study using data from NHANES

Xiaoqin Qu[1,2], Jingcheng Jiang 🆔[3]*

1 Medical Imaging Center, the Second People's Hospital of Yibin, Yibin, China, 2 Clinical Research and Translational Center, Neuroimaging Big Data Research Center, the Second People's Hospital of Yibin, Yibin, China, 3 Department of Neurosurgery, the Second People's Hospital of Yibin, Yibin, China

* jingchengjiang1985@163.com

## Abstract

### Background

Osteoporosis, a skeletal disorder characterized by reduced bone density and increased fracture risk, imposes a significant global health burden, particularly in aging populations. Previous studies have highlighted the negative impact of alcohol consumption on bone health, but the interplay between osteoporosis and mortality risk in alcohol-consuming individuals remains underexplored. This study aimed to investigate the association between osteoporosis and cardiovascular disease (CVD) mortality and all-cause mortality in U.S. adults who consume alcohol.

### Methods

This prospective cohort study utilized data from the National Health and Nutrition Examination Survey (NHANES) spanning five cycles (2005–2010, 2013–2014, and 2017–2018). A total of 12,178 alcohol-consuming participants aged 20 years and older were included after excluding those with missing data or non-drinking status. Bone density was measured using dual-energy X-ray absorptiometry (DXA), and osteoporosis was defined using World Health Organization (WHO) T-score criteria (T-score ≤ −2.5). Mortality data were obtained through linkage with the National Death Index (NDI). Multivariable Cox proportional hazards regression models were employed to assess the relationship between osteoporosis and mortality outcomes, adjusting for demographic, socioeconomic, and clinical covariates.

### Results

Kaplan-Meier survival analysis revealed higher all-cause and CVD mortality rates in participants with osteoporosis compared to those without (Log-rank test P < 0.001

**Data availability statement:** The data underlying the results presented in the study are available from (https://www.cdc.gov/nchs/nhanes/index.htm).

**Funding:** The author(s) received no specific funding for this work.

**Competing interests:** The authors have declared that no competing interests exist. All authors have no conflicts of interest.

**Abbreviations:** CVD, Cardiovascular Disease; OP, Osteoporosis; NHANES, National Health and Nutrition Examination Survey; MEC, Mobile Examination Center; DXA, dual-energy X-ray absorptiometry; who, World Health Organization; BMD, bone mineral density; NDI, National Death Index; ICD-10, International Classification of Diseases (ICD-10); BMI, body mass index; CI, confidence interval; PIR, Poverty income ratio; CHD, coronary heart disease; CHF, Congestive Heart Failure; DM, diabetes mellitus; GLU, Fasting glucose; ALT, alanine aminotransferase; AST, Aspartate transaminase; TBil, Total Bilirubin; Alb, serum albumin; GGT, glutamyl transpeptidase; CRE, Blood creatinine; UA, Blood uric acid; BUN, blood urea nitrogen; Na, Serum sodium; P, Blood phosphorus; Ca: serum calcium; K, Serum potassium; Fe, serum iron; Cl, Serum chlorine; TG, Triglyceride; TC, triglyceride; HDL, high-density lipoprotein; LDL, low-density lipoprotein; HR, Hazard ratios.

for both). After adjusting for potential confounders, osteoporosis was associated with a 1.60-fold increased risk of all-cause mortality (HR = 1.60, 95% CI = 1.26–2.03, P < 0.001) and a 1.55-fold increased risk of CVD mortality (HR = 1.55, 95% CI = 1.06–2.28, P = 0.025). Stratified analyses across age, sex, smoking status, and cardiovascular risk factors showed consistent results, with no significant interaction effects (P > 0.05). Sensitivity analyses confirmed the robustness of these findings.

## Conclusion

Osteoporosis is positively associated with increased risks of all-cause and CVD mortality in alcohol-consuming individuals. These findings underscore the need for further research to elucidate the underlying mechanisms and inform preventive strategies targeting this high-risk population.

## Introduction

Osteoporosis is a disease characterized by reduced bone density and impaired microarchitecture, leading to increased skeletal fragility and elevated fracture risk [1]. It imposes a significant societal and economic burden globally, particularly in elderly populations [2]. Studies indicate that as populations age, the number of osteoporosis-related fractures and associated healthcare costs are projected to increase. By 2025, direct medical expenditures attributed to osteoporosis are estimated to reach approximately $25.3 billion [3].

The relationship between alcohol consumption and osteoporosis has garnered significant attention. Research indicates that chronic excessive alcohol consumption may negatively impact bone health, leading to reduced bone density and increased fracture risk [4]. The mechanisms through which alcohol affects bone health are complex and may involve interference with bone formation and resorption, as well as effects on the endocrine system [5]. Moreover, alcohol consumption may exacerbate the risk of osteoporosis by impairing nutrient absorption and metabolism [6]. Beyond individual health, alcohol consumption poses a threat to societal well-being. Alcohol-related health issues, such as cardiovascular diseases, liver diseases, cancers, and mental and neurological disorders, contribute to healthcare costs and productivity losses, imposing a heavy socioeconomic burden [7,8]. Studies have highlighted that alcohol is the third leading risk factor for global disease burden, following smoking and hypertension [9].

Previous studies have explored the relationship between alcohol consumption and osteoporosis, but most have focused on the direct impact of alcohol on bone density, neglecting the complex interplay between alcohol intake and osteoporosis [10]. To clarify the exact relationship between these risk factors, further research must be conducted. This will enable the implementation of effective preventive measures to reduce disease risk. Therefore, a large – scale cohort study involving this specific demographic, along with a more in – depth review of the impact on CVD mortality and all – cause mortality risk, is needed. In this study, we used the National Health

and Nutrition Examination Survey (NHANES) to assess the relationship between osteoporosis and CVD and all – cause mortality risk in alcohol – consuming individuals in the United States.

## Method

### Study population

Since 1999, the NHANES has collected and analyzed health and nutrition data of the U.S. population to meet diverse information needs [11]. NHANES includes comprehensive health examinations, laboratory tests, and dietary interviews, covering participants of all ages. These data provide essential evidence for the development and evaluation of public health policies [12]. Through these interviews and tests, NHANES offers valuable data to understand the complex relationship among diet, nutrition, and health, and provides a scientific basis for public health policy – making [13].

### Identification of drinking population

NHANES covers alcohol consumption in individuals aged 20 and older over the past 12 months, with variables prefixed as "ALQ". Alcohol consumption categories are based on frequency and quantity [14]. Personal interviews using NHANES CAPI software are conducted at Mobile Examination Center(MEC). Participants are asked if they consume at least 12 alcoholic drinks per year. If they answer affirmatively, further questions are asked about drinking days per week, month, or year, resulting in five alcohol consumption statuses: (1) lifetime consumption of fewer than 12 drinks, (2) ≥12 drinks in a year but no consumption in the past year, or no consumption in the past year but lifetime consumption of ≥12 drinks, (3) current light drinking: ≤ 1 drink/day for women, ≤ 2 drinks/day for men, (4) current moderate drinking: ≥ 2 drinks/day for women, ≥ 3 drinks/day for men, or ≥2 drinking days/month, (5) current heavy drinking: ≥ 3 drinks/day for women, ≥ 4 drinks/day for men, or ≥5 days/month of binge drinking [≥4 drinks on an occasion for women, ≥ 5 for men]. The latter three statuses are defined as alcohol – consuming individuals [15]. We categorised alcohol consumption into three groups: light drinkers (≤1 drink per day for women and ≤2 drinks per day for men), moderate drinkers (≥2 drinks per day for women and ≥3 drinks per day for men, or ≥2 days of heavy drinking per month), and heavy drinkers (≥3 drinks per day for women and ≥4 drinks per day for men, or heavy drinking for ≥5 days per month). Heavy drinking was defined as ≥4 drinks on a single occasion for women and ≥5 drinks on a single occasion for men.

### Bone density measurement and definition of osteoporosis

Since 2005, NHANES has performed dual-energy X-ray absorptiometry (DXA) scans of the proximal femur for eligible survey participants aged 8 and older at the MEC [16]. Pregnant women were excluded from DXA examinations. DXA scans were performed using a Hologic QDR-4500A fan-beam densitometer (Hologic, Inc., Bedford, Massachusetts). The radiation exposure from femoral DXA scans is extremely low, less than 20 microsieverts.

According to the World Health Organization (WHO) criteria, T-scores are calculated by comparing an individual's bone mineral density (BMD) with the average BMD of healthy young adults [17]. This method plays a significant role in the diagnosis of osteoporosis. In this study, T-scores ≤ −2.5 were defined as osteoporosis, while T-scores between −1.0 and −2.5 were classified as non-osteoporotic [17].

### Determination of death data

NHANES mortality and follow-up data are obtained by matching with the National Death Index (NDI), a resource that helps researchers determine if study participants have died [18]. NDI also provides selected mortality information for participants' deaths, including death dates and causes, as well as International Classification of Diseases (ICD-10) codes [19]. In this study, mortality status was updated through April 28, 2022.

We collected data from 28,470 participants aged 20 and older across five cycles of NHANES (2005–2010, 2013–2014, and 2017–2018). Other cycles were excluded due to unavailable data on total femur and femoral neck bone mineral

density. We excluded 11,890 participants due to missing alcohol – related data or non – drinking status, 4,387 due to missing osteoporosis data, and 15 due to incomplete mortality follow – up. A total of 12,178 participants were included (see Fig 1). The NCHS Research Ethics Review Board ensured informed consent from all participants. For detailed statistics, refer to the National Health and Nutrition Examination Survey website (https://www.cdc.gov/nchs).

## Covariant

We referenced a series of comprehensive factors from prior studies as covariates [20]. Demographic factors included age, sex, and BMI, while sociocultural factors were represented by race, education, family income, marital status, and smoking status. BMI was categorized into three groups: 18.5 to 25 kg/m², 25–30 kg/m², and ≥30 kg/m². Race/ethnicity was captured by dividing participants into three groups: non-Hispanic White, non-Hispanic Black, and other (multiracial). Educational attainment was categorized as less than high school, high school diploma or equivalent, and college graduate or higher. Marital status was defined as cohabiting (married or living with a partner) versus living alone (widowed, divorced, separated, or never married). Family income, based on the poverty-income ratio, was stratified into three categories: ≤ 1.30, 1.31–3.50, and >3.5. Smoking status was divided into two groups: non-smokers (<100 cigarettes in a lifetime) and former smokers (>100 cigarettes in a lifetime but currently abstinent) were classified as non-smokers, while current smokers (>100 cigarettes in a lifetime and still smoking) were classified as smokers. We also considered physician-diagnosed conditions, including coronary heart disease, stroke, congestive heart failure, hyperlipidemia, hypertension, and diabetes, to highlight the profound impact of chronic diseases on our study.

We collected various blood biochemical indicators, including fasting serum glucose, alanine aminotransferase, aspartate aminotransferase, total bilirubin, serum albumin, gamma-glutamyl transferase, serum creatinine, serum uric acid, blood urea nitrogen, serum sodium, serum phosphorus, serum calcium, serum potassium, serum iron, serum chloride, triglycerides, total cholesterol, high-density lipoprotein, and low-density lipoprotein. The NHANES website provides detailed descriptions of blood sample collection, storage, transportation, and laboratory procedures.

## Data statistical methods

Continuous variables were presented as mean (standard error, SE), and categorical variables as weighted percentages for descriptive statistics. Group comparisons were performed using chi – square tests and analysis of variance. Missing data for covariates were addressed using multiple imputation.

Multivariable Cox proportional hazards regression models were used to evaluate all – cause and CVD mortality risks. Hazard ratios (HR) and corresponding 95% confidence intervals (CI) for the exposed group were calculated, with non – osteoporotic participants as the reference. Covariates were adjusted in three steps: Model 1 was unadjusted. Model 2 adjusted for age, sex, race, marital status, family income and education, smoking, BMI, coronary heart disease, stroke, congestive heart failure, hyperlipidemia, hypertension, and diabetes. Model 3 further adjusted for fasting serum glucose, alanine aminotransferase, aspartate aminotransferase, total bilirubin, serum albumin, gamma – glutamyl transferase, serum creatinine, serum uric acid, blood urea nitrogen, serum sodium, phosphorus, calcium, potassium, iron, chloride, triglycerides, total cholesterol, high – density lipoprotein, and low – density lipoprotein, based on Model 2.

Kaplan – Meier survival curves were plotted for survival analysis, with intergroup differences assessed by the log – rank test. Stratified analyses were performed on subgroups defined by age, sex, smoking status, hyperlipidaemia, hypertension, and diabetes, as well as on alcohol – consumption subgroups categorised into light, moderate, and heavy drinking, to evaluate potential differences among these subgroups.

In the propensity score matching (PSM) analysis, patients with osteoporosis were 1:1 matched to healthy controls using the nearest neighbour method. During the matching process, all covariates were adjusted for as confounding factors (S1 Fig). Subsequently, the matched data underwent repeat COX multivariate analysis to verify result stability. For covariates with missing values of less than 5%, multiple imputation by chained equations (assuming data were missing

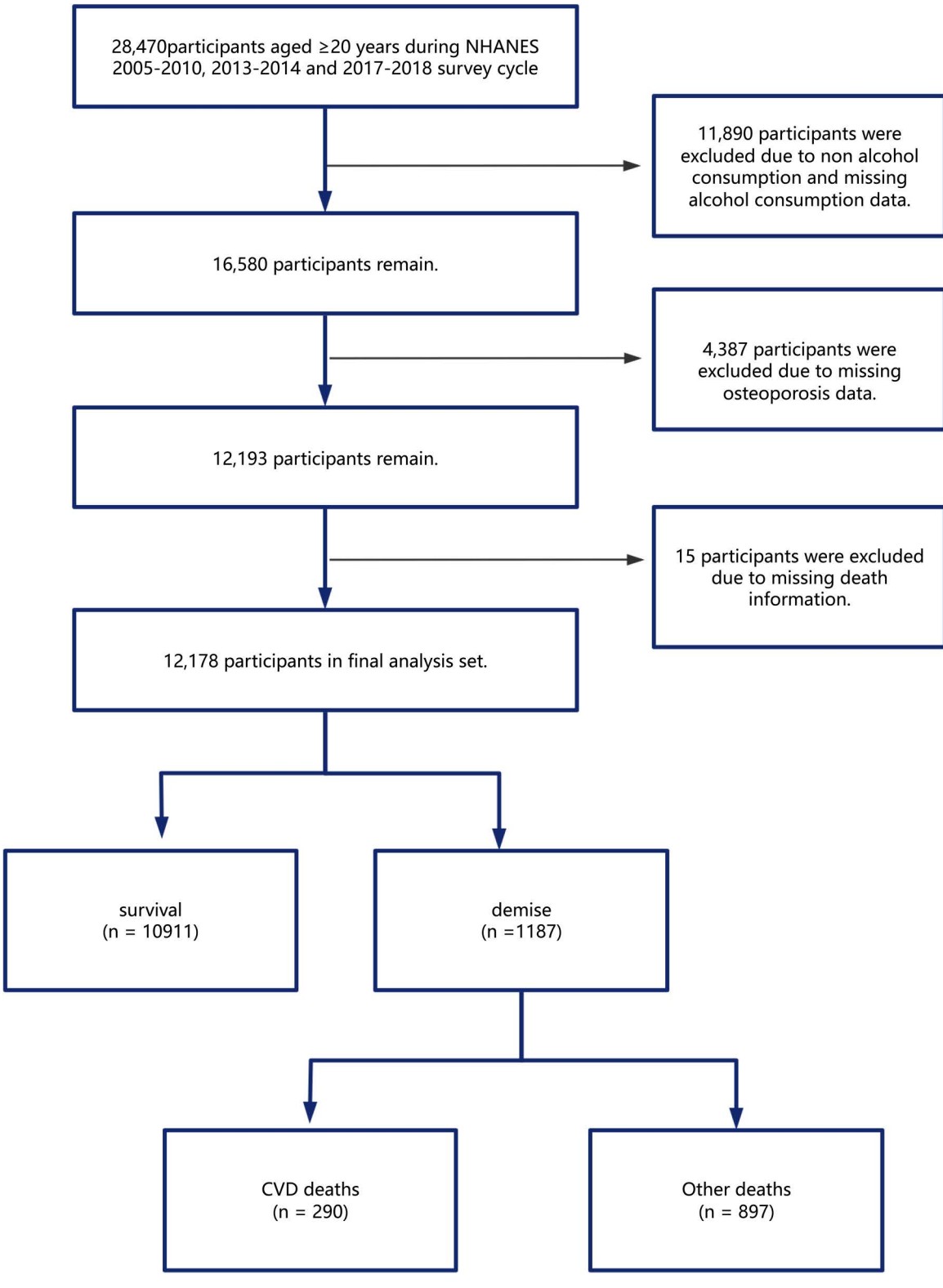

**Fig 1. Flow chart of patient inclusion.**

at random, MAR) was applied. Five imputed datasets were generated, including all analysis variables and outcome indicators.

All data were analyzed using R (version 4.2.2, The R Foundation) and the Free Statistics analysis platform (version 2.0, Beijing, China). A p-value of less than 0.05 was considered statistically significant.

## Result

### Baseline characteristic

Our analysis included 12,178 alcohol-consuming participants aged 20 and older, with a weighted total of 107,882,689. Of these, 596 (weighted at 5,108,640) had osteoporosis, and 12,178 (weighted at 102,774,049) did not. Baseline data for both groups are in Table 1. Osteoporosis individuals tended to be female, non-Hispanic White, and older. They often had lower BMI and were less likely to smoke. Compared with non-osteoporosis subjects, those with osteoporosis had higher rates of hyperlipidemia but lower rates of hypertension and diabetes.

### Osteoporosis and mortality rate

Among the 12,178 participants with a median follow – up of 123 months and a mean follow – up of 111 ± 47 months.Kaplan-Meier curves showed higher all-cause mortality in osteoporosis participants than in non-osteoporosis participants (Log-rank test P < 0.001, Fig 2). In multivariable Cox proportional hazards models, after adjusting for potential confounders, osteoporosis was associated with a 1.60 – fold higher risk of all – cause mortality (HR = 1.60, 95%CI = 1.26–2.03, P < 0.001) compared with non – osteoporosis (Table 2, Model 3).

Our results also showed a significantly higher risk of CVD mortality in participants with osteoporosis than in those without (Log-rank test P < 0.001, Fig 2). In the multivariable Cox proportional hazards model adjusted for potential confounders, participants with osteoporosis had a 1.55 – fold higher risk of CVD mortality compared to those without (HR = 1.55, 95%CI = 1.06–2.28, P = 0.025) (Table 2, Model 3) (Fig 3).

In the multivariable Cox proportional hazards model of the propensity – matched data, after adjusting for potential confounding factors, osteoporosis was associated with a 1.32 – fold increased risk of all – cause mortality compared to non – osteoporosis (HR = 1.32, 95% CI = 1.03–1.67, P = 0.026). However, osteoporosis was not significantly associated with CVD mortality compared to non – osteoporosis (S1 Table).

### Stratification and sensitivity analysis

Stratified analyses of subgroups (age, sex, smoking status, hyperlipidaemia, hypertension, diabetes) showed no interactions (P for interaction>0.05, Figs 4 and 5). Similarly, analyses stratified by alcohol – consumption levels (light, moderate, heavy) indicated robust results (P for interaction>0.05, Fig 6). Sensitivity analyses revealed no significant differences between included and excluded participants. Even after excluding patients with missing values, the link between osteoporosis and all – cause/CVD mortality remained significant.

## Discuss

In this large prospective cohort study of alcohol consumers, we observed a positive correlation between osteoporosis and risks of all – cause and CVD mortality, with HRs of 1.60 (95%CI = 1.26–2.03) for all – cause mortality and 1.55 (95%CI = 1.06–2.28) for CVD mortality. The significant associations between osteoporosis and event – based all – cause/CVD mortality were consistent across subgroups defined by age, smoking status, and CVD risk factors (diabetes, hypertension, hyperlipidemia). Sensitivity analyses also confirmed the robustness of the results.

Our findings are consistent with prior studies [21,22]. First, individuals with osteoporosis often have comorbidities such as malnutrition and chronic diseases, which may increase mortality risk [23]. Additionally, alcohol consumption itself may negatively affect bone density, worsening the severity of osteoporosis and further elevating mortality [22].

**Table 1. Baseline characteristics (weighted) grouped based on the presence of osteoporosis.**

| Characteristic | Overall, N = 107,882,689 | OP-, N = 102,774,049 | OP +, N = 5,108,640 | p-value[1] |
|---|---|---|---|---|
| Age(mean.std.error) | 48.72 (0.30) | 47.98 (0.29) | 63.66 (0.80) | <0.001 |
| Sex, n (%) | | | | <0.001 |
| Male | 6,695 (52.28%) | 6,530 (53.67%) | 165 (24.23%) | |
| Female | 5,483 (47.72%) | 5,052 (46.33%) | 431 (75.77%) | |
| Race, n (%) | | | | <0.001 |
| Non-Hispanic White | 6,083 (73.80%) | 5,710 (73.35%) | 373 (83.05%) | |
| Non-Hispanic Black | 2,287 (9.29%) | 2,241 (9.59%) | 46 (3.31%) | |
| Other Race | 3,808 (16.91%) | 3,631 (17.07%) | 177 (13.64%) | |
| Marry, n (%) | | | | <0.001 |
| Married | 7,596 (66.23%) | 7,286 (66.85%) | 310 (53.71%) | |
| Never married | 4,582 (33.77%) | 4,296 (33.15%) | 286 (46.29%) | |
| PIR, n (%) | | | | 0.001 |
| ≤1.30 | 3,069 (15.77%) | 2,913 (15.70%) | 156 (17.16%) | |
| 1.31-3.50 | 4,484 (33.19%) | 4,229 (32.75%) | 255 (41.89%) | |
| >3.5 | 4,625 (51.05%) | 4,440 (51.55%) | 185 (40.95%) | |
| EDU, n (%) | | | | 0.60 |
| Less than high school | 2,613 (13.29%) | 2,489 (13.34%) | 124 (12.18%) | |
| High school or equivalent | 2,848 (23.23%) | 2,697 (23.12%) | 151 (25.47%) | |
| Above high school | 6,717 (63.48%) | 6,396 (63.54%) | 321 (62.35%) | |
| smoke, n (%) | | | | 0.23 |
| no | 9,184 (77.12%) | 8,722 (77.24%) | 462 (74.53%) | |
| yes | 2,994 (22.88%) | 2,860 (22.76%) | 134 (25.47%) | |
| BMI, n (%) | | | | <0.001 |
| 18.5~24.99 kg/m2 | 3,625 (31.49%) | 3,272 (29.97%) | 353 (62.01%) | |
| 25.00~29.9 kg/m2 | 4,400 (34.98%) | 4,234 (35.45%) | 166 (25.63%) | |
| ≥30.00 kg/m2 | 4,153 (33.53%) | 4,076 (34.58%) | 77 (12.36%) | |
| CHD, n (%) | | | | 0.005 |
| no | 11,691 (96.58%) | 11,140 (96.72%) | 551 (93.73%) | |
| yes | 487 (3.42%) | 442 (3.28%) | 45 (6.27%) | |
| stroke, n (%) | | | | <0.001 |
| no | 11,797 (97.57%) | 11,242 (97.76%) | 555 (93.92%) | |
| yes | 381 (2.43%) | 340 (2.24%) | 41 (6.08%) | |
| CHF, n (%) | | | | 0.001 |
| no | 11,882 (98.29%) | 11,316 (98.37%) | 566 (96.63%) | |
| yes | 296 (1.71%) | 266 (1.63%) | 30 (3.37%) | |
| Hyperlipidemia, n (%) | | | | 0.033 |
| no | 3,474 (28.72%) | 3,351 (28.98%) | 123 (23.35%) | |
| yes | 8,704 (71.28%) | 8,231 (71.02%) | 473 (76.65%) | |
| Hypertension, n (%) | | | | <0.001 |
| no | 7,109 (62.17%) | 6,842 (62.70%) | 267 (51.49%) | |
| yes | 5,069 (37.83%) | 4,740 (37.30%) | 329 (48.51%) | |
| DM, n (%) | | | | 0.76 |
| no | 10,272 (88.03%) | 9,765 (88.01%) | 507 (88.57%) | |
| yes | 1,906 (11.97%) | 1,817 (11.99%) | 89 (11.43%) | |

*(Continued)*

**Table 1.** (Continued)

| Characteristic | Overall, N = 107,882,689 | OP-, N = 102,774,049 | OP +, N = 5,108,640 | p-value¹ |
|---|---|---|---|---|
| GLU.mmol/L | 5.85 (0.02) | 5.85 (0.02) | 5.75 (0.06) | 0.80 |
| Alt.U/L | 25.93 (0.21) | 26.12 (0.21) | 22.04 (0.85) | <0.001 |
| Ast.U/L | 25.71 (0.18) | 25.67 (0.17) | 26.60 (1.43) | 0.33 |
| TBil.umol/L | 12.44 (0.09) | 12.51 (0.09) | 11.22 (0.41) | <0.001 |
| Alb.g/L | 42.67 (0.06) | 42.71 (0.06) | 41.91 (0.17) | <0.001 |
| GGT.U/L | 29.78 (0.45) | 29.80 (0.46) | 29.41 (2.25) | 0.002 |
| CRE.umol/L | 79.85 (0.29) | 79.96 (0.30) | 77.64 (1.33) | <0.001 |
| UA.umol/L | 325.87 (1.01) | 327.52 (1.06) | 292.60 (3.70) | <0.001 |
| BUN.mmol/L | 4.79 (0.03) | 4.77 (0.03) | 5.24 (0.12) | <0.001 |
| Na. mmol/L | 139.41 (0.08) | 139.39 (0.08) | 139.79 (0.14) | <0.001 |
| P. mmol/L | 1.21 (0.00) | 1.21 (0.00) | 1.24 (0.01) | 0.019 |
| Ca. mmol/L | 2.36 (0.00) | 2.36 (0.00) | 2.36 (0.01) | 0.40 |
| K. mmol/L | 4.01 (0.01) | 4.00 (0.01) | 4.08 (0.02) | 0.006 |
| Fe. µmol/L | 16.20 (0.09) | 16.20 (0.10) | 16.17 (0.34) | 0.85 |
| Cl. mmol/*L* | 103.58 (0.08) | 103.61 (0.08) | 102.81 (0.19) | <0.001 |
| TG. mmol/L | 2.58 (0.03) | 2.58 (0.03) | 2.56 (0.13) | 0.96 |
| TC. mmol/L | 5.10 (0.01) | 5.09 (0.01) | 5.24 (0.06) | 0.059 |
| HDL. mmol/L | 1.41 (0.01) | 1.40 (0.01) | 1.65 (0.02) | <0.001 |
| LDL. mmol/L | 2.54 (0.01) | 2.54 (0.01) | 2.45 (0.07) | 0.10 |
| Year, n (%) | | | | <0.001 |
| 2005-2006 | 2,422 (23.47%) | 2,330 (23.80%) | 92 (16.69%) | |
| 2007-2008 | 3,043 (23.78%) | 2,942 (24.35%) | 101 (12.21%) | |
| 2009-2010 | 3,383 (24.79%) | 3,250 (25.10%) | 133 (18.52%) | |
| 2013-2014 | 1,951 (16.40%) | 1,812 (16.00%) | 139 (24.39%) | |
| 2017-2018 | 1,379 (11.57%) | 1,248 (10.75%) | 131 (28.20%) | |
| all_cause_mort, n (%) | 1,187 (6.95%) | 1,027 (6.21%) | 160 (21.89%) | <0.001 |
| cvd_mort, n (%) | 290 (1.60%) | 251 (1.44%) | 39 (4.83%) | <0.001 |

¹ Wilcoxon rank-sum test for complex survey samples; chi-squared test with Rao & Scott's second-order correction.

OP-: No osteoporosis,OP +: osteoporosis,PIR: Poverty income ratio, EDU: education, BMI: Body Mass Index, CHD: coronary heart disease, CHF: Congestive Heart Failure,DM: diabetes mellitus, GLU: Fasting glucose, ALT: alanine aminotransferase, AST: Aspartate transaminase,TBil: *Total Bilirubin,Alb: serum albumin,GGT: glutamyl transpeptidase,* CRE: Blood creatinine,UA: Blood uric acid, BUN: blood urea nitrogen,Na: Serum sodium,P: Blood phosphorus,Ca: serum calcium,K: Serum potassium,Fe: serum iron,Cl: Serum chlorine,TG: Triglyceride,TC: triglyceride,HDL: high-density lipoprotein,LDL: low-density lipoprotein.

The mechanisms underlying the synergistic effects of osteoporosis and alcohol consumption on cardiovascular disease (CVD) and all – cause mortality risk are multidimensional and interactive [24]. At the biological level, acetaldehyde, a metabolite of alcohol, inhibits osteoblast activity and activates osteoclasts, reducing bone density. Simultaneously, the oxidative stress response induced by acetaldehyde impairs vascular endothelial function and promotes atherosclerotic plaque formation [25,26]. Notably, alcohol – induced disruption of vitamin D metabolism exacerbates bone calcium loss and increases cardiac afterload by up – regulating the renin – angiotensin system, creating a vicious cycle of the "bone - vascular calcification axis"[27].

From a sociological perspective, long – term alcohol consumers often exhibit imbalanced dietary patterns (e.g., low calcium and high sodium intake) and poor exercise adherence [28]. These factors, combined with osteoporosis, form a "social exposome" of metabolic syndrome. Chronic release of inflammatory cytokines accelerates arterial stiffness [29,30].

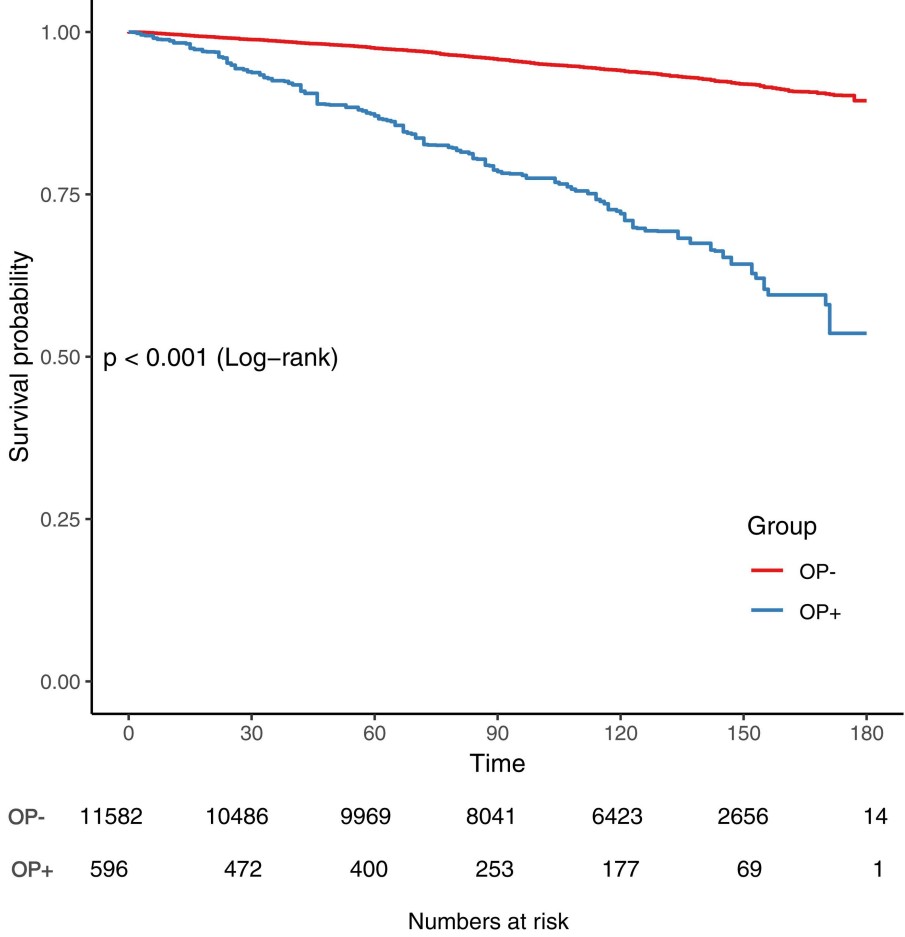

OP-    11582       10486        9969         8041         6423         2656          14

OP+     596          472          400          253          177           69            1

Numbers at risk

**Fig 2. Osteoporosis and all-cause mortality survival curve (weighted).**

**Table 2. Association of osteoporosis with all-cause mortality and CVD mortality in the alcohol consuming population (weighted).**

| Variable | Model 1 | | Model 2 | | Model 3 | |
|---|---|---|---|---|---|---|
| | HR(95%CI) | P_value | HR(95%CI) | P_value | HR(95%CI) | P_value |
| **All-cause mortality** | | | | | | |
| OP- | Reference | | Reference | | Reference | |
| OP+ | 5.47(4.41-6.78) | <0.001 | 1.71(1.35-2.16) | <0.001 | 1.60(1.26-2.03) | <0.001 |
| **CVD mortality** | | | | | | |
| OP- | Reference | | Reference | | Reference | |
| OP+ | 5.19(3.70-7.27) | <0.001 | 1.61(1.11-2.35) | 0.013 | 1.55(1.06-2.28) | 0.025 |

Model 1: Adjustment.

Model 2: adjusted for age (continuous), race and ethnicity (non Hispanic white, non Hispanic black, Mexican American, other Hispanics, other/multiracial), education level (lower than high school, high school graduation or equivalent, college graduation or above), smoking status (smokers, non-smokers), drinking status (drinkers, non-smokers) BMI (18.5–24.99 kg/m2, 25–29.99 kg/m2, ≥ 30 kg/m2), coronary heart disease, stroke, congestive heart failure, hyperlipidemia, hypertension, diabetes.

Model 3: Further adjust serum glucose, alanine aminotransferase, aspartate aminotransferase, total bilirubin, serum albumin, glutamine transpeptidase, serum creatinine, serum uric acid, blood urea nitrogen, serum sodium ions, serum phosphorus, serum calcium ions, serum potassium ions, serum iron ions, serum chloride ions, triglycerides, total cholesterol, high-density lipoprotein, and low-density lipoprotein based on Model 2.

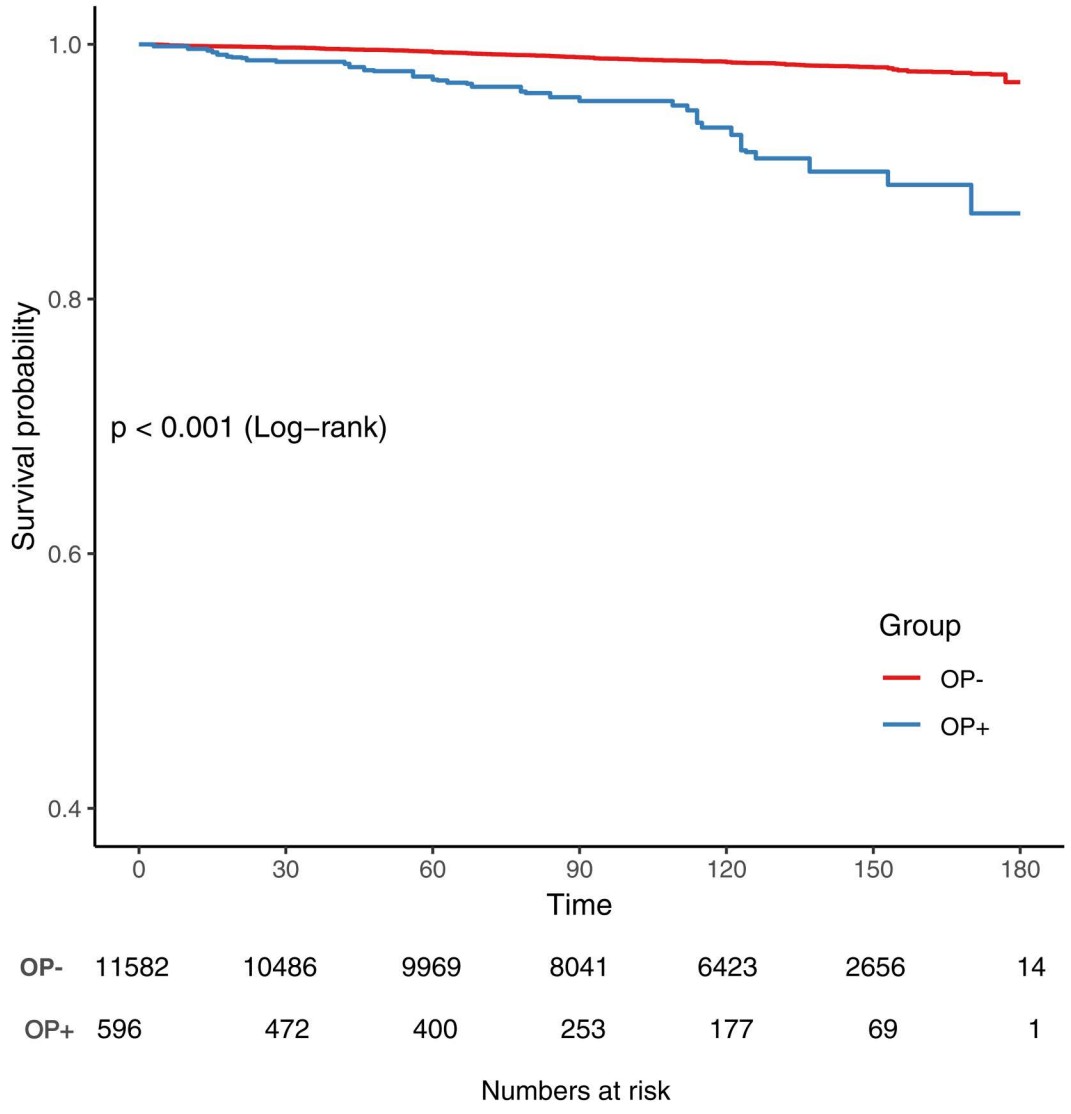

**Fig 3. Osteoporosis and CVD mortality Survival Curve (Weighted).**

Clinically, elevated serum bone resorption markers in osteoporosis patients not only reflect bone metabolic disorders but also correlate positively with vascular calcification markers, suggesting that bone – derived factors may directly participate in vascular remodeling via paracrine pathways [31,32].

Furthermore, alcohol – induced activation of hepatic cytochrome P450 enzymes accelerates the metabolism of anti – osteoporosis drugs, leading to treatment resistance. Bone pain and other clinical symptoms may prompt patients to increase alcohol consumption, forming a behavioral feedback loop [33,34]. These multi – system interactions ultimately result in endothelial dysfunction, accelerated cardiac remodeling, and reduced immune surveillance, manifesting as elevated mortality risk.

This study, based on NHANES data of alcohol – consuming individuals, provides clear insights into the relationship between osteoporosis and the risks of all – cause and CVD mortality. Its major strengths lie in the large sample size, population – based design, and the ability to examine osteoporosis alongside all – cause and CVD mortality and

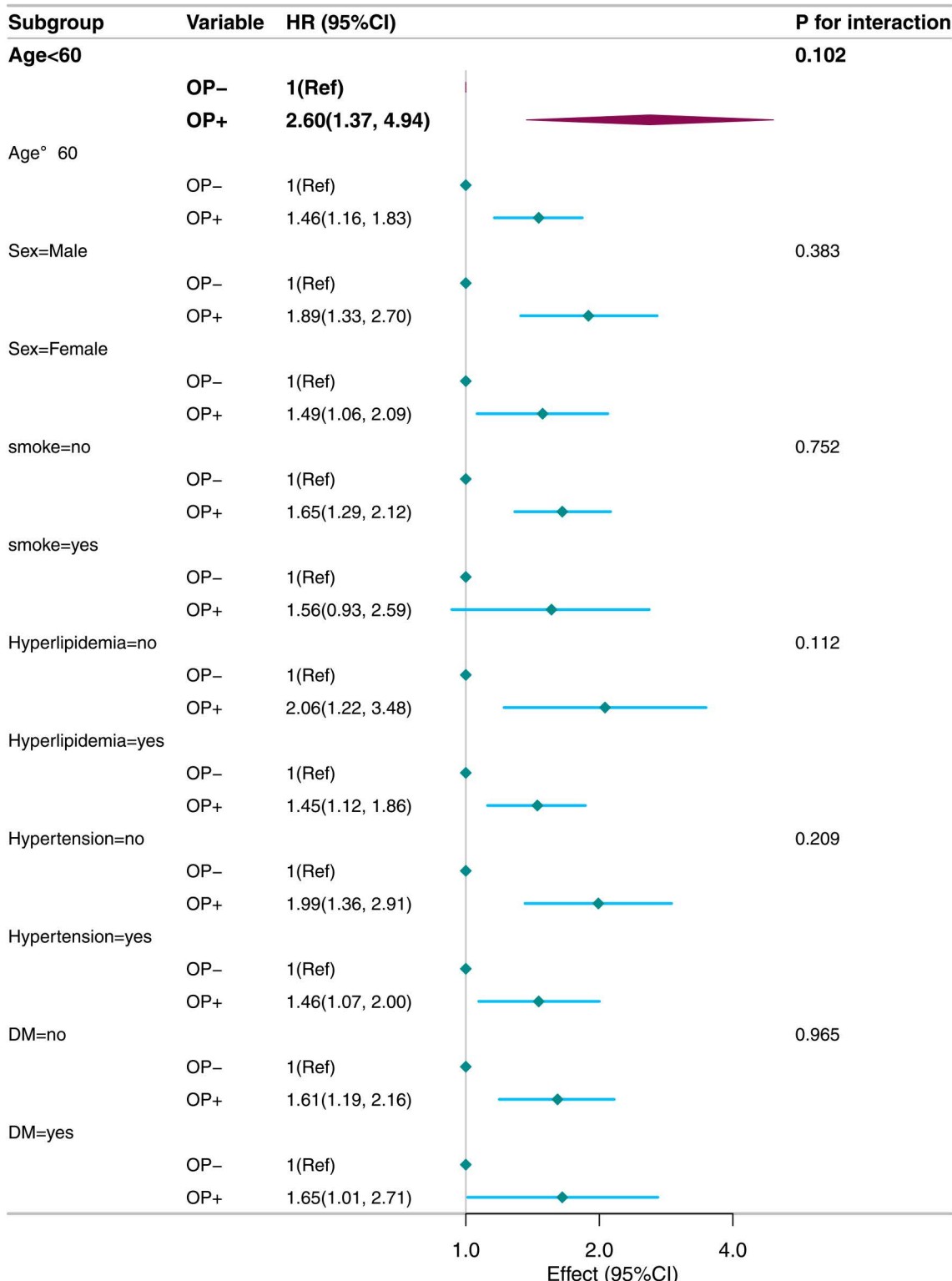

| Subgroup | Variable | HR (95%CI) | P for interaction |
|---|---|---|---|
| **Age<60** | | | **0.102** |
| | **OP−** | **1(Ref)** | |
| | **OP+** | **2.60(1.37, 4.94)** | |
| Age° 60 | | | |
| | OP− | 1(Ref) | |
| | OP+ | 1.46(1.16, 1.83) | |
| Sex=Male | | | 0.383 |
| | OP− | 1(Ref) | |
| | OP+ | 1.89(1.33, 2.70) | |
| Sex=Female | | | |
| | OP− | 1(Ref) | |
| | OP+ | 1.49(1.06, 2.09) | |
| smoke=no | | | 0.752 |
| | OP− | 1(Ref) | |
| | OP+ | 1.65(1.29, 2.12) | |
| smoke=yes | | | |
| | OP− | 1(Ref) | |
| | OP+ | 1.56(0.93, 2.59) | |
| Hyperlipidemia=no | | | 0.112 |
| | OP− | 1(Ref) | |
| | OP+ | 2.06(1.22, 3.48) | |
| Hyperlipidemia=yes | | | |
| | OP− | 1(Ref) | |
| | OP+ | 1.45(1.12, 1.86) | |
| Hypertension=no | | | 0.209 |
| | OP− | 1(Ref) | |
| | OP+ | 1.99(1.36, 2.91) | |
| Hypertension=yes | | | |
| | OP− | 1(Ref) | |
| | OP+ | 1.46(1.07, 2.00) | |
| DM=no | | | 0.965 |
| | OP− | 1(Ref) | |
| | OP+ | 1.61(1.19, 2.16) | |
| DM=yes | | | |
| | OP− | 1(Ref) | |
| | OP+ | 1.65(1.01, 2.71) | |

Effect (95%CI)

**Fig 4. Stratified analysis of osteoporosis and all-cause mortality (weighted).**

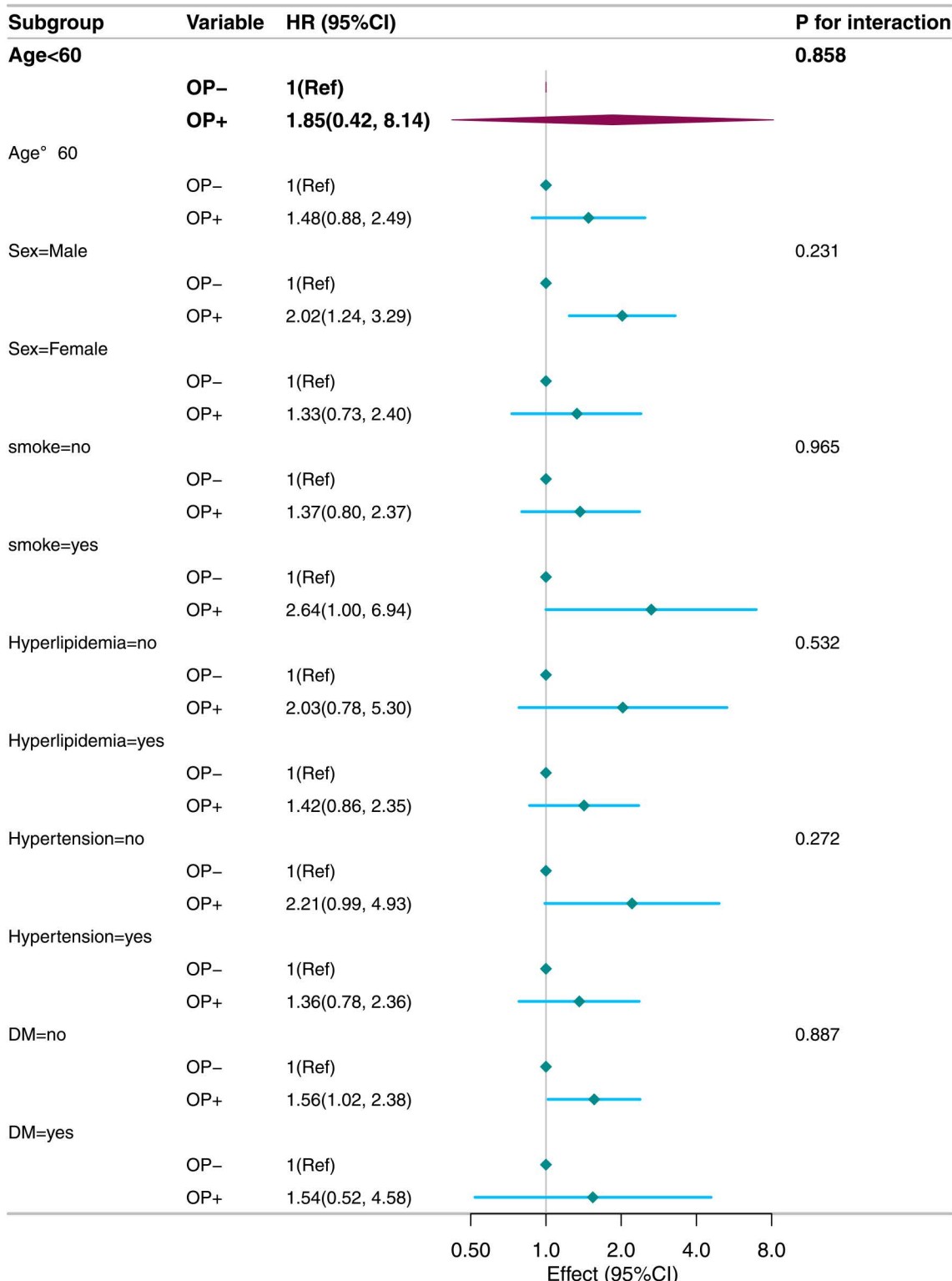

| Subgroup | Variable | HR (95%CI) | | P for interaction |
|---|---|---|---|---|
| **Age<60** | | | | **0.858** |
| | **OP−** | **1(Ref)** | | |
| | **OP+** | **1.85(0.42, 8.14)** | | |
| Age° 60 | | | | |
| | OP− | 1(Ref) | | |
| | OP+ | 1.48(0.88, 2.49) | | |
| Sex=Male | | | | 0.231 |
| | OP− | 1(Ref) | | |
| | OP+ | 2.02(1.24, 3.29) | | |
| Sex=Female | | | | |
| | OP− | 1(Ref) | | |
| | OP+ | 1.33(0.73, 2.40) | | |
| smoke=no | | | | 0.965 |
| | OP− | 1(Ref) | | |
| | OP+ | 1.37(0.80, 2.37) | | |
| smoke=yes | | | | |
| | OP− | 1(Ref) | | |
| | OP+ | 2.64(1.00, 6.94) | | |
| Hyperlipidemia=no | | | | 0.532 |
| | OP− | 1(Ref) | | |
| | OP+ | 2.03(0.78, 5.30) | | |
| Hyperlipidemia=yes | | | | |
| | OP− | 1(Ref) | | |
| | OP+ | 1.42(0.86, 2.35) | | |
| Hypertension=no | | | | 0.272 |
| | OP− | 1(Ref) | | |
| | OP+ | 2.21(0.99, 4.93) | | |
| Hypertension=yes | | | | |
| | OP− | 1(Ref) | | |
| | OP+ | 1.36(0.78, 2.36) | | |
| DM=no | | | | 0.887 |
| | OP− | 1(Ref) | | |
| | OP+ | 1.56(1.02, 2.38) | | |
| DM=yes | | | | |
| | OP− | 1(Ref) | | |
| | OP+ | 1.54(0.52, 4.58) | | |

Effect (95%CI): 0.50  1.0  2.0  4.0  8.0

**Fig 5. Stratified analysis of osteoporosis and CVD mortality (weighted).**

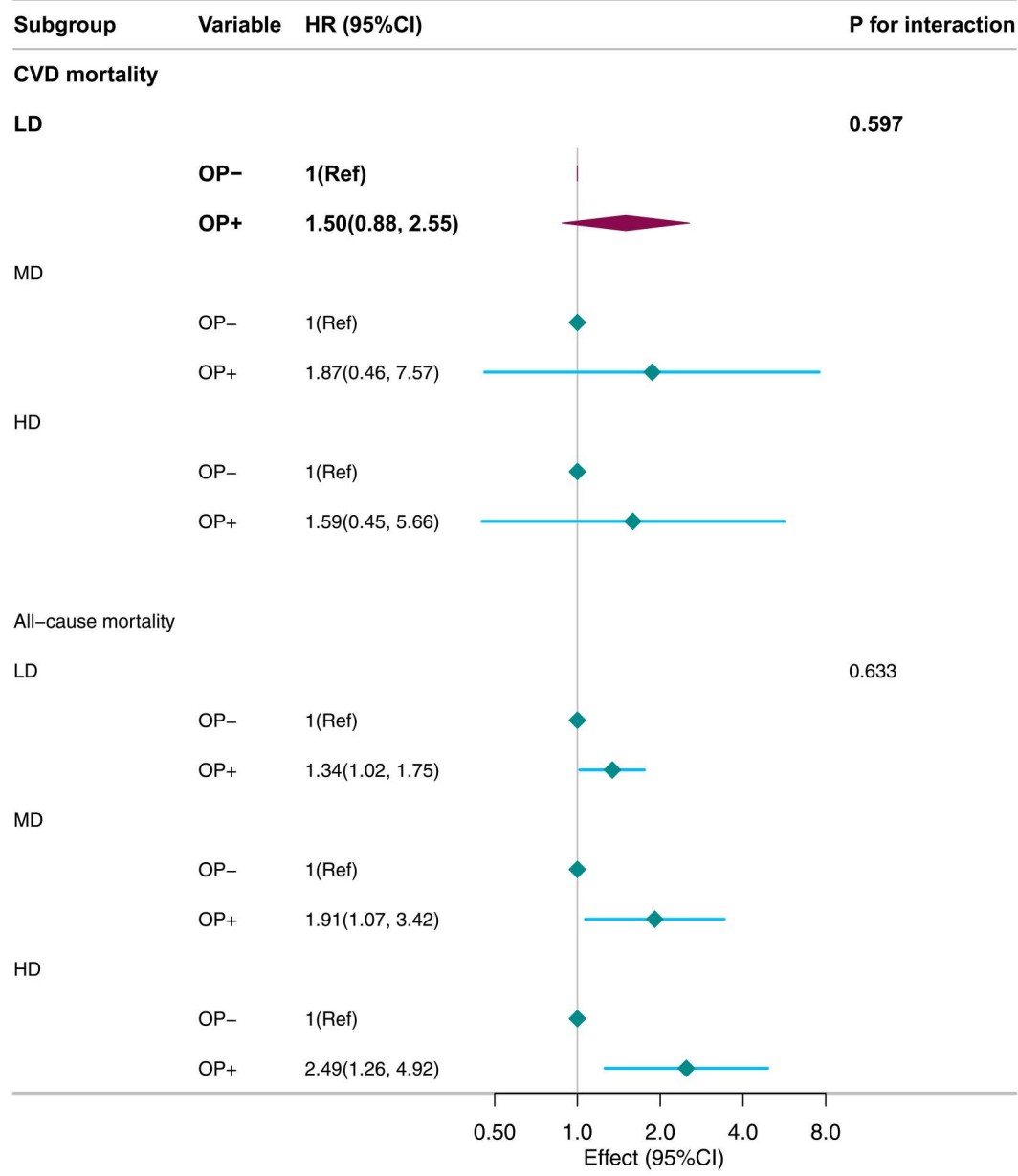

**Fig 6. Stratified analysis of mortality rate based on alcohol consumption level.**

subgroup – related risks. As a prospective cohort study with strict inclusion and exclusion criteria, it adjusted for various confounders, including demographic, chronic disease, and biochemical indicators, offering a unique perspective with great novelty and clinical significance. However, as an observational cohort study, the design cannot exclude the possibility of reverse causation or residual confounding factors. Participants with missing alcohol – consumption data, non – drinkers, and those with missing osteoporosis data were excluded, which introduced selection bias. Further in – depth studies are needed in view of these limitations. like integrating DXA scans into alcohol cessation programs for clearer guidance to clinicians and policymakers.

Osteoporosis is positively correlated with all – cause and CVD mortality risks in alcohol – consuming individuals. These findings warrant further attention and research to better understand the health impacts of osteoporosis.

## Supporting information

**S1 Fig. Standardized mean difference plot after propensity score matching.**
(PDF)

**S1 Table. Association between osteoporosis and CVD and all-cause mortality after propensity score matching.**
(DOCX)

## Acknowledgments

We thank all the participants who volunteered as part of the NHANES. We thank the Free Statistics team for providing technical assistance and valuable tools for data analysis and visualization. The authors acknowledge Jie Liu of the Department of Vascular and Endovascular Surgery, Chinese PLA General Hospital, Huanxian Liu, Department of Neurology, Chinese PLA General Hospital, for his contribution to statistical support, study design consultations, and comments regarding the manuscript.

## Author contributions

**Conceptualization:** Jingcheng Jiang, Xiaoqin Qu.

**Data curation:** Jingcheng Jiang, Xiaoqin Qu.

**Formal analysis:** Jingcheng Jiang, Xiaoqin Qu.

**Funding acquisition:** Jingcheng Jiang, Xiaoqin Qu.

**Investigation:** Jingcheng Jiang, Xiaoqin Qu.

**Methodology:** Jingcheng Jiang, Xiaoqin Qu.

**Project administration:** Jingcheng Jiang, Xiaoqin Qu.

**Resources:** Jingcheng Jiang, Xiaoqin Qu.

**Software:** Jingcheng Jiang, Xiaoqin Qu.

**Supervision:** Jingcheng Jiang, Xiaoqin Qu.

**Validation:** Jingcheng Jiang, Xiaoqin Qu.

**Visualization:** Jingcheng Jiang, Xiaoqin Qu.

**Writing – original draft:** Jingcheng Jiang, Xiaoqin Qu.

**Writing – review & editing:** Jingcheng Jiang, Xiaoqin Qu.

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
