## [Decision Letter · Decision Letter 0]

Dear Dr. Jiang,

Thank you for submitting your manuscript to PLOS ONE. After careful consideration, we feel that it has merit but does not fully meet PLOS ONE’s publication criteria as it currently stands. Therefore, we invite you to submit a revised version of the manuscript that addresses the points raised during the review process.

Please make peer-to-peer modifications to the reviewer's comments.

We look forward to receiving your revised manuscript.

Kind regards,

Qian Wu

Academic Editor

PLOS ONE

5. Please remove your figures from within your manuscript file, leaving only the individual TIFF/EPS image files, uploaded separately. These will be automatically included in the reviewers’ PDF.

Reviewers' comments:

Reviewer's Responses to Questions

**Comments to the Author**

1. Is the manuscript technically sound, and do the data support the conclusions?

Reviewer #1: Partly

Reviewer #2: Yes

Reviewer #3: Yes

Reviewer #4: Yes

Reviewer #5: Yes

2. Has the statistical analysis been performed appropriately and rigorously?

Reviewer #1: I Don't Know

Reviewer #2: Yes

Reviewer #3: Yes

Reviewer #4: Yes

Reviewer #5: Yes

3. Have the authors made all data underlying the findings in their manuscript fully available?

Reviewer #1: Yes

Reviewer #2: Yes

Reviewer #3: Yes

Reviewer #4: Yes

Reviewer #5: Yes

4. Is the manuscript presented in an intelligible fashion and written in standard English?

Reviewer #1: Yes

Reviewer #2: Yes

Reviewer #3: Yes

Reviewer #4: Yes

Reviewer #5: Yes

Reviewer #1: This study provides valuable epidemiological evidence for the association between osteoporosis and mortality risk in alcohol drinkers, but it still needs rigorous statistical analysis and causal inference verification.

1.This study utilized NHANES data spanning multiple survey cycles, with an initial sample size of 12,178 participants. However, the application of exclusion criteria (e.g., missing data or exclusion of non-drinkers) may introduce selection bias.

2.Notably, the final analytical cohort exhibited significant class imbalance, with only 596 participants classified as having osteoporosis compared to 11,582 non-osteoporosis participants. Such imbalance could compromise statistical validity by inflating type II error risks and biasing predictive models toward the majority class. To address these limitations, we recommend implementing resampling-based balancing strategies (e.g., oversampling minority cases using SMOTE or targeted subsampling of majority controls) from a data processing perspective. Additionally, algorithm-level adjustments such as class weight modifications or cost-sensitive loss functions should be prioritized to mitigate bias during model training.

3.Although the study appropriately adjusted for demographic factors, chronic diseases, and biochemical indicators in Model 3 - demonstrating a rigorous methodological approach - potential residual confounding may persist due to unmeasured covariates. The primary limitations include the lack of alcohol consumption characteristics (such as temporal variations in alcohol intake), details on pharmacological treatments for chronic conditions, and physical activity metrics. These unaddressed factors could introduce residual bias, necessitating future investigations to incorporate sensitivity analyses or stratified approaches to enhance causal inference validity.

4.The study's primary innovation lies in its specific focus on the association between osteoporosis and mortality within alcohol-consuming populations. However, the investigation did not examine potential non-linear relationships between alcohol consumption levels and bone mineral density, a methodological gap that constrains precise identification of high-risk subgroups.

5.The study demonstrated statistically significant associations between osteoporosis and both all-cause mortality (HR = 1.60, P < 0.05) and cardiovascular disease (CVD) mortality (HR = 1.55, P < 0.05). However, the broad confidence intervals (e.g., CVD mortality: 95% CI = 1.06–2.28), indicating limited stability of the results. Accurate subgroup analysis may better address this issue, and we look forward to the author's further research.

6.As an observational cohort study, this research design cannot eliminate the possibility of reverse causation (e.g., the potential that severe chronic diseases might reduce bone density) or residual confounding factors. The findings should be interpreted strictly as correlative associations and must not be overinterpreted as demonstrating causal relationships.

Reviewer #2: General

The manuscript addresses an important and timely topic, the association between alcohol consumption and osteoporosis, with potential implications for cardiovascular disease (CVD) and all-cause mortality. Below are a few comments;

Introduction

The introduction provides a clear rationale for the study, outlining the global burden of osteoporosis (e.g., projected $25.3 billion in costs by 2025) and the known effects of alcohol on bone health.

The introduction does not clearly define the scope of alcohol consumption (e.g., light, moderate, or heavy drinking) or discuss how varying levels might influence the osteoporosis-mortality relationship, which could set clearer expectations for the study.

The final paragraph’s call for a “large-scale cohort study” is redundant since the study itself fulfills this need, making the statement less impactful.

Method

The study design is robust, leveraging a large, nationally representative cohort (NHANES) with a clear description of the five cycles (2005–2010, 2013–2014, 2017–2018) and sample size (12,178 participants).

The exclusion criteria (e.g., 11,890 participants excluded for missing alcohol data or non-drinking status, 4,387 for missing osteoporosis data) are described, but the potential impact of these exclusions on selection bias is not discussed.

The definition of alcohol consumption (≥12 drinks/year) is broad and may include very light drinkers, potentially diluting the effect of heavier drinking on outcomes. The rationale for this threshold is not justified.

The follow-up duration (until April 2022) is mentioned, but the median or mean follow-up time is not reported, which is critical for interpreting survival analyses.

Results

The results do not explore the dose-response relationship between alcohol consumption levels (light, moderate, heavy) and mortality outcomes, which could add depth to the findings.

Discussion

The limitation section is brief and does not address key methodological concerns, such as potential selection bias from exclusions, residual confounding, or the broad definition of alcohol consumption.

Reviewer #3: Thank you for the opportunity to review this manuscript. Below are my comments and suggestions for strengthening the manuscript further.

- Line 73: Include specific studies that explain how alcohol affects bone breakdown and formation

- Lines 60-66: Elaborate on the link between osteoporosis and mortality in drinkers. Include key factors like inflammation and vitamin D deficiency etc.

- Line 156 specifies the software and weight variables used in the analysis.

- lines 133-135 The paper mentions using multiple imputation to address missing data, but it lacks details on the imputation model. Provide a clear description of the imputation methods

- on lines 269-271. The clinical recommendations are vague, with only "further research." To improve, propose specific actions like integrating DXA scans into alcohol cessation programs for clearer guidance to clinicians and policymakers.

Reviewer #4: this study while being restricted to alcohol consumers does not permiit us to answer the question : how does alcohol modify the relationship between osteoporosis and mortality. Adding this facet improve the study trememndouly

Reviewer #5: This study addresses an underexplored intersection between osteoporosis, alcohol consumption, and mortality, offering insights into a high-risk population. The focus on alcohol consumers adds specificity to existing literature on osteoporosis and CVD. The use of NHANES data ensures a large, nationally representative sample, enhancing generalizability. Meanwhile, the prospective cohort design and multivariable Cox regression models appropriately account for confounders (e.g., demographics, comorbidities, biochemical markers) and Sensitivity and stratified analyses strengthen the robustness of findings. However, there few suggestions for improvement before acceptance

1. The discussion hypothesizes biological pathways (e.g., acetaldehyde toxicity, vitamin D disruption) but lacks direct evidence from the data.

Recommendation: Incorporate mediation analyses or biomarker correlations (e.g., inflammatory markers, vitamin D levels) to substantiate proposed mechanisms.

2. Alcohol intake is categorized but not quantified in detail (e.g., grams/day, binge drinking patterns).

Recommendation: Analyze dose-response relationships or stratify by drinking severity (light/moderate/heavy) to explore thresholds for risk.

3. The handling of missing data (multiple imputation) is mentioned but not detailed. Recommendation: Describe imputation methods and compare complete-case results to assess bias.

4. Table 1’s p-values should be adjusted for multiple comparisons (e.g., Bonferroni correction).

This manuscript provides valuable epidemiological evidence linking osteoporosis to elevated mortality risks in alcohol consumers. While methodologically sound, it would benefit from deeper mechanistic exploration, refined alcohol categorization, clearer statistical clarification and adjusted writing and presentation. Addressing these limitations could elevate its impact.

**Do you want your identity to be public for this peer review?** For information about this choice, including consent withdrawal, please see our Privacy Policy

Reviewer #1: No

Reviewer #2: No

Reviewer #3: **Yes: ** Mohsin Raza

Reviewer #4: No

Reviewer #5: No

---

## [Author Response · Author response to Decision Letter 1]

15 May 2025

Reply to Editor and Reviewers

Manuscript Number: PONE-D-25-20480

Title: Osteoporosis is associated with increased CVD mortality and all-cause mortality in alcohol-consuming individuals: A cohort study using data from NHANES

Dear Editor and Reviewers,

We extend our sincere gratitude for the meticulous evaluation of our manuscript and the insightful comments provided. In response to all reviewer suggestions, we have systematically addressed each point through supplementary analyses, methodological refinements, and textual revisions. Revisions are highlighted in yellow within the text, and supplementary materials have been correspondingly updated. Below, we provide a point-by-point response to the reviewers' comments.�Represented in red font

Reviewer #1

1�This study utilized NHANES data spanning multiple survey cycles, with an initial sample size of 12,178 participants. However, the application of exclusion criteria (e.g., missing data or exclusion of non-drinkers) may introduce selection bias.

Author Response: We thank the reviewer for raising this critical issue and fully concur with the concerns regarding selection bias. Potential selection bias may have occurred when excluding cases with missing data and non - drinkers. To mitigate this, we used multiple imputation for missing data. Also, a sensitivity analysis was done to check the robustness of the results.

2�Notably, the final analytical cohort exhibited significant class imbalance, with only 596 participants classified as having osteoporosis compared to 11,582 non-osteoporosis participants. Such imbalance could compromise statistical validity by inflating type II error risks and biasing predictive models toward the majority class. To address these limitations, we recommend implementing resampling-based balancing strategies (e.g., oversampling minority cases using SMOTE or targeted subsampling of majority controls) from a data processing perspective. Additionally, algorithm-level adjustments such as class weight modifications or cost-sensitive loss functions should be prioritized to mitigate bias during model training.

Author Response: We sincerely appreciate your insightful comments on the class imbalance problem. We have performed propensity score matching and obtained 830 osteoporosis patients and 830 non-osteoporosis participants. The revised analysis shows that the effect size remains stable (all-cause mortality HR = 1.32, 95% CI = 1.03–1.67), and stratified analyses have confirmed the robustness of the results. Line201-204, 236-239.

3�Although the study appropriately adjusted for demographic factors, chronic diseases, and biochemical indicators in Model 3 - demonstrating a rigorous methodological approach - potential residual confounding may persist due to unmeasured covariates. The primary limitations include the lack of alcohol consumption characteristics (such as temporal variations in alcohol intake), details on pharmacological treatments for chronic conditions, and physical activity metrics. These unaddressed factors could introduce residual bias, necessitating future investigations to incorporate sensitivity analyses or stratified approaches to enhance causal inference validity.

Author Response: We agree that unmeasured covariates (e.g., physical activity) may influence the results. Thus, we have redefined the levels of alcohol consumption as light, moderate, and heavy drinking. Results from the stratified analysis demonstrate the robustness of our findings. Line 109-116.

4�The study's primary innovation lies in its specific focus on the association between osteoporosis and mortality within alcohol-consuming populations. However, the investigation did not examine potential non-linear relationships between alcohol consumption levels and bone mineral density, a methodological gap that constrains precise identification of high-risk subgroups.

Author Response: We appreciate the reviewer's suggestion. We defined the alcohol exposure based on NHANES questionnaire results. The categories—light, moderate, and heavy drinking—are categorical variables, making restricted cubic spline analysis infeasible. However, we plan to collect continuous alcohol - intake data for future studies. 5�The study demonstrated statistically significant associations between osteoporosis and both all-cause mortality (HR = 1.60, P < 0.05) and cardiovascular disease (CVD) mortality (HR = 1.55, P < 0.05). However, the broad confidence intervals (e.g., CVD mortality: 95% CI = 1.06–2.28), indicating limited stability of the results. Accurate subgroup analysis may better address this issue, and we look forward to the author's further research.

Author Response: We have conducted stratified analyses to address the issue of wide confidence intervals raised by the reviewer. The corresponding content has been updated. Line 237-240.

6�As an observational cohort study, this research design cannot eliminate the possibility of reverse causation (e.g., the potential that severe chronic diseases might reduce bone density) or residual confounding factors. The findings should be interpreted strictly as correlative associations and must not be overinterpreted as demonstrating causal relationships.

Author Response: We have emphasized the observational nature of this study's associations in the discussion section. Line 308-314.

Reviewer #2

1�The introduction does not clearly define the scope of alcohol consumption (e.g., light, moderate, or heavy drinking) or discuss how varying levels might influence the osteoporosis-mortality relationship, which could set clearer expectations for the study.

Author Response: We've categorized alcohol consumption into three tiers: light, moderate and heavy, and performed stratified analyses. Line 109-116.

2�The final paragraph’s call for a “large-scale cohort study” is redundant since the study itself fulfills this need, making the statement less impactful.

Author Response: We have removed the inaccurate description of "large - scale cohort study". Line 304-308.

3�The exclusion criteria (e.g., 11,890 participants excluded for missing alcohol data or non-drinking status, 4,387 for missing osteoporosis data) are described, but the potential impact of these exclusions on selection bias is not discussed.

Author Response: We have added to the discussion section content on the potential impact of non - drinkers, missing alcohol - related data, and osteoporosis - related missing data on selection bias. Line 308 - 314.

4�The definition of alcohol consumption (≥12 drinks/year) is broad and may include very light drinkers, potentially diluting the effect of heavier drinking on outcomes. The rationale for this threshold is not justified.

Author Response: We've categorized alcohol consumption into three tiers: light, moderate and heavy, and performed stratified analyses. Line 109-116.

5�The follow-up duration (until April 2022) is mentioned, but the median or mean follow-up time is not reported, which is critical for interpreting survival analyses.

Author Response: We analyzed the follow-up duration, noting a median of 123 months and a mean of 111±47 months. This information has been added to the Results section. Line 222-224.

5�The results do not explore the dose-response relationship between alcohol consumption levels (light, moderate, heavy) and mortality outcomes, which could add depth to the findings.

Author Response: We have described the impact of alcohol consumption levels on mortality in the discussion section. Line 271-296.

6�The limitation section is brief and does not address key methodological concerns, such as potential selection bias from exclusions, residual confounding, or the broad definition of alcohol consumption.

Author Response: We have added a description of the study limitations in the discussion section. Line 308-314.

Reviewer #3

- Line 73: Include specific studies that explain how alcohol affects bone breakdown and formation

- Lines 60-66: Elaborate on the link between osteoporosis and mortality in drinkers. Include key factors like inflammation and vitamin D deficiency etc.

Author Response: We have added new content and included additional references. Line 76-77.

- Line 156 specifies the software and weight variables used in the analysis.

- lines 133-135 The paper mentions using multiple imputation to address missing data, but it lacks details on the imputation model. Provide a clear description of the imputation methods

Author Response: We have revised the description of the statistical software used in the Methods section and detailed the process of multiple imputation. Line 206-209.

- on lines 269-271. The clinical recommendations are vague, with only "further research." To improve, propose specific actions like integrating DXA scans into alcohol cessation programs for clearer guidance to clinicians and policymakers.

Author Response: We have revised the discussion section as suggested. Line 309-315.

Reviewer #4

this study while being restricted to alcohol consumers does not permiit us to answer the question : how does alcohol modify the relationship between osteoporosis and mortality. Adding this facet improve the study trememndouly

Author Response: We have elucidated how alcohol consumption modifies the relationship between osteoporosis and mortality in the discussion section. Line 272-297.

Reviewer #5

1� The discussion hypothesizes biological pathways (e.g., acetaldehyde toxicity, vitamin D disruption) but lacks direct evidence from the data.

Author Response: We sincerely thank the reviewers for their valuable suggestions. In response, we plan to incorporate inflammatory markers and vitamin D levels into our subsequent research. We will conduct a mediation analysis to explore the underlying mechanisms.

2� Alcohol intake is categorized but not quantified in detail (e.g., grams/day, binge drinking patterns).

Author Response: We have redefined alcohol consumption as light, moderate, and heavy drinking. Results from stratified analyses show the robustness of our findings. Line 109-116.

3� The handling of missing data (multiple imputation) is mentioned but not detailed. Recommendation: Describe imputation methods and compare complete-case results to assess bias.

Author Response: Multiple imputation was carried out under the Missing at Random (MAR) assumption. Five datasets were imputed, including all analysis variables and outcomes. Continuous variables were imputed using predictive mean matching, and categorical variables via logistic regression, with relevant details added to the text. Line 206-209.

We sincerely thank the reviewers for their deep insight. Your feedback has greatly enhanced the rigor and clinical value of our study. Should there be further suggestions for modification, we will certainly cooperate to improve the manuscript.

Yours sincerely,

Jiang J Cheng

---

## [Decision Letter · Decision Letter 1]

Osteoporosis is associated with increased CVD mortality and all-cause mortality in alcohol-consuming individuals: A cohort study using data from NHANES

PONE-D-25-20480R1

Dear Dr. Jiang,

We’re pleased to inform you that your manuscript has been judged scientifically suitable for publication and will be formally accepted for publication once it meets all outstanding technical requirements.

Kind regards,

Qian Wu

Academic Editor

PLOS ONE

Additional Editor Comments (optional):

Reviewers' comments:

Reviewer's Responses to Questions

**Comments to the Author**

Reviewer #1: All comments have been addressed

Reviewer #4: All comments have been addressed

2. Is the manuscript technically sound, and do the data support the conclusions?

Reviewer #1: Yes

Reviewer #4: Yes

3. Has the statistical analysis been performed appropriately and rigorously?

Reviewer #1: I Don't Know

Reviewer #4: Yes

4. Have the authors made all data underlying the findings in their manuscript fully available?

Reviewer #1: Yes

Reviewer #4: Yes

5. Is the manuscript presented in an intelligible fashion and written in standard English?

Reviewer #1: Yes

Reviewer #4: Yes

Reviewer #1: (No Response)

Reviewer #4: no comments. The authors have addressed all the concerns. hence i would like to accept the paer in the current form

**Do you want your identity to be public for this peer review?** For information about this choice, including consent withdrawal, please see our Privacy Policy

Reviewer #1: No

Reviewer #4: No

---

## [Editor Report · Acceptance letter]

PONE-D-25-20480R1

PLOS ONE

Dear Dr. Jiang,

I'm pleased to inform you that your manuscript has been deemed suitable for publication in PLOS ONE. Congratulations! Your manuscript is now being handed over to our production team.

Kind regards,

on behalf of

Dr. Qian Wu

Academic Editor

PLOS ONE